# Navigating the Challenges of Gluten Enteropathy and Infertility: The Role of Celiac-Related Antibodies and Dietary Changes

**DOI:** 10.3390/antib12040079

**Published:** 2023-12-06

**Authors:** Monika Peshevska-Sekulovska, Milena Gulinac, Radoslav Rangelov, Desislava Docheva, Tsvetelina Velikova, Metodija Sekulovski

**Affiliations:** 1Medical Faculty, Sofia University St. Kliment Ohridski, Kozyak 1 Str., 1407 Sofia, Bulgaria; monika.p.sekulovska@gmail.com (M.P.-S.); tsvelikova@medfac.mu-sofia.bg (T.V.); 2Department of Gastroenterology, University Hospital Lozenetz, 1407 Sofia, Bulgaria; 3Department of General and Clinical Pathology, Medical University of Plovdiv, 15A Vasil Aprilov Bul. 4000 Plovdiv, Bulgaria; mgulinac@hotmail.com; 4Medical Center Neovitro OOD, 20 Petko Yu. Todorov Bul., 1408 Sofia, Bulgaria; r.rangelov@neovitro.bg (R.R.); docheva@neovitro.bg (D.D.); 5Department of Anesthesiology and Intensive Care, University Hospital Lozenetz, 1 Kozyak Str., 1407 Sofia, Bulgaria

**Keywords:** celiac disease, gluten enteropathy, autoantibodies, anti-tTG, anti-EMA, gluten, reproduction failure, infertility, miscarriage

## Abstract

Celiac disease (CD) is an autoimmune condition that is initiated in genetically susceptible individuals by the exposure of the intestines to gluten, and the early start of symptoms is related to malabsorption. Atypical variants of the illness are often identified in adulthood and are frequently associated with manifestations outside of the intestines, including metabolic osteopathy, anemia, and dermatitis herpetiformis. But also, empirical data suggest a correlation between CD and reproductive abnormalities, including repeated abortions. Infertility and repeated miscarriages frequently manifest in women diagnosed with CD and may serve as the initial clinical indication of a subclinical form. Furthermore, the condition may manifest as amenorrhea, infertility, and the delivery of infants with a low birth weight. Regarding the mechanisms of CD in infertility, along with the anti-tTG action to hinder the invasiveness of trophoblast, these antibodies could damage endometrial angiogenesis, which has been shown in in vitro models with human endometrial cells and in vivo in murine models. Another important aspect is the role of nutrient deficiencies, such as zinc deficiency (connected to impaired hormone production, secondary amenorrhea, and pre-eclampsia) and folic acid, etc. Therefore, our objective was to conduct a comprehensive review of the existing literature pertaining to this specific topic and to elucidate the role of the autoantibodies in its pathogenesis.

## 1. Introduction

Celiac disease (CD) is an autoimmune condition that is initiated in genetically susceptible individuals by the exposure of the intestines to gluten, a protein that is typically present in wheat, barley, and rye [1]. The normal presentation of CD is characterized by an early start of symptoms that are related to malabsorption. In contrast, atypical variants of the illness are often identified in adulthood and are frequently associated with manifestations outside of the intestines. CD has been linked to many extra-intestinal manifestations, including metabolic osteopathy, anemia, and dermatitis herpetiformis [2]. Empirical data also suggest a correlation between CD and reproductive abnormalities, including repeated abortions. Infertility and repeated miscarriages frequently manifest in women diagnosed with CD and may serve as the initial clinical indication of a subclinical form. Furthermore, the condition may manifest as amenorrhea, infertility, repeated abortions, and the delivery of infants with a low birth weight [3]. Approximately 0.5% to 1% of the general population is affected by celiac disease (CD), with a notable gender bias towards females (female/male ratio: 3/1). These figures display considerable fluctuations across distinct geographical regions [4]. A study conducted in 2003 unveiled a CD prevalence of 6.25% among patients grappling with “idiopathic” infertility [1].

Chronic inflammation has the potential to generate certain serological markers that could potentially influence the progression of infertility [2]. The utilization of serological markers in screening procedures has revealed that there has been a previous lack of accurate diagnosis for CD and that the disease’s prevalence in the global population is estimated to be at least 1 in 250, if not more. Transglutaminase 2 (TG2) serves as the primary autoantigen and plays a significant role as a pathogenetic factor in the development of the disease. It has been proven that anti-tTG 2 has been interfering with throphoblast, leading to miscarriage or low-birth-weight neonates. Consequently, it has garnered considerable attention as a potential target for research efforts aimed at developing novel therapeutic approaches [5]. This gives rise to the question of whether GFD would be a possible cure for infertility in CD patients?

In light of the nature of dietary habits, it is important to underline that they significantly impact overall health, and emerging evidence indicates that they can similarly influence fertility and reproductive outcomes in all genders. The researchers discovered that while women diagnosed with CD exhibited normal fertility on the whole, there was a decline in fertility levels over the final two years prior to the diagnosis of CD. This highlights the significance of untreated CD as a risk factor for infertility and suggests that a gluten-free diet (GFD), one of the available therapeutic strategies, may effectively rectify this anomaly [6]. Moreover, as observed in the case of other disorders affecting organs beyond the intestines, the timely implementation of a GFD has the potential to reduce the likelihood of developing this particular issue.

A literature search was conducted in the databases Medline (PubMed) and Scopus using MeSH and relevant free-text terms: (“celiac disease” AND “infertility”) AND (“celiac disease” AND “reproduction”), (“Celiac disease” OR “Gluten enteropathy”) (“infertility” OR “reproductive problems”). References were further hand-searched for supplements (Figure 1). The association between CD and infertility has been examined in many studies, yielding inconsistent findings. Therefore, our objective was to conduct a comprehensive review of the existing literature pertaining to reproductive problems and CD. We also attempt to elucidate the role of the autoantibodies in its pathogenesis to explain the exact immunological mechanisms of gluten-related autoantibodies and their role in infertility and dietary habits, their association with infertility, and last but not least GFD and its effect on the nutritional status of pregnant women, as well as its effect on serological, endoscopic, and histological markers.

## 2. Infertility and Celiac Disease

There is increasing evidence that CD can affect fertility in both men and women. The most common manifestation of infertility in men due to CD is impotence and reduced sexual activity, but hypogonadism and impaired sperm parameters are also observed. A seminal analysis of coeliacs revealed marked abnormalities of sperm morphology and motility, but only the former appeared to improve after gluten withdrawal [7]. A study by Ludvigsson et al. reports that babies whose fathers have CD have lower birth weights compared with controls. On average, these babies weighed 266 g less. Additionally, infants whose fathers had CD experienced a shorter pregnancy duration compared with other infants. The percentage of preterm births for infants whose fathers had coeliac disease was higher and almost significant, with an adjusted odds ratio of 3.32 [8]. Another interesting point was presented in a study by Sahin et al. They found out that in the siblings of celiac patients, the prevalence of CD was 10.7%. A proportion of 33.3% of the siblings who received a diagnosis of CD exhibited no symptoms. HLA-DQ alleles were identified in 98.2% of individuals diagnosed with celiac disease, and in 100% of siblings who were also diagnosed with celiac disease. Furthermore, one of the two siblings received a diagnosis of CD one year subsequent to the first diagnosis, while the other sibling was diagnosed four years later [9].

On the other hand, women with celiac disease experience serious menstrual problems, including significantly delayed menarche, more frequent secondary amenorrhea, and a tendency towards early menopause. These complications may be the first symptom of CD [10]. The high risk of complications is reported in several other studies as well. In the study by Smecuol et al., celiac patients had twice as many abortions compared with controls (*p* < 0.02) [9], and in the study by Moltini [10], the frequency of recurrent abortions was also significantly higher (*p* < 0.03). Martinelli et al. studied 845 women attending an obstetric clinic and found 12 women with previously unidentified, minimally symptomatic CD, 7 of whom had poor outcomes due to low birth weight or spontaneous abortion [11]. In accordance with that observation, the frequency of low-birth-weight babies is much higher in patients with untreated CD compared with control populations. Mothers bearing the HLA-DQ2 or -DQ8 molecule exhibited a two-fold increase in the impact of intermediate anti-tTG levels on birth weight. There is evidence to suggest that impaired fetal growth development is linked to metabolic changes that may predispose individuals to the development of adult illnesses, including cardiovascular disease, hypertension, and type 2 diabetes mellitus. Furthermore, there is evidence to suggest that individuals with a low birth weight exhibit impaired cognitive functioning [12].

Numerous hypotheses have been postulated in order to elucidate the correlation between celiac disease and birth outcomes. It is widely recognized that the nutritional state of mothers, specifically deficiencies in iron, folate, and vitamin B12, as well as the maternal body mass index (BMI) during pregnancy, have a significant impact on the growth of the fetus [13]. Furthermore, recent research has suggested that anti-tTG may have a separate and distinct impact on the formation and functioning of the placenta. Furthermore, it was shown that the human placenta exhibited the presence of tTG expression. Simone et al. demonstrated that anti-tTG antibodies elicit death in trophoblast cells, which play a critical role in placental development [14,15].

Existing data revealed that the frequency of these serious complications is not linked to the severity of CD, and the risks are significantly reduced after treatment with a GFD. The duration of breastfeeding is also affected after birth and is 2.5 times shorter in untreated patients [16,17]. There are concerns about unveiling previously undiagnosed CD during pregnancy and postpartum. Different authors report cases where CD was diagnosed for the first time after childbirth. Malnick et al. report three cases where previously healthy women developed diarrhea, weight loss, and malabsorption after delivery and were subsequently diagnosed with CD [18]. Similarly, Corrado et al. report ten cases of CD being diagnosed after pregnancy [19]. The emergence of new autoimmune diseases during pregnancy and an early postpartum period are not uncommon findings and have been described earlier with respect to rheumatoid arthritis, systemic lupus erythematosus, and other connective tissue disorders [20,21].

Such an association has been thought to be due to higher levels of sex hormones during pregnancy and their pronounced impact on the immune system [18].

**Along this line of reasoning, CD may also be associated with prolonged oxidative stress. The role of oxidative stress in subclinical forms of the disease is highlighted by Odetti et al.** [21]**. In their study, the levels of markers of oxidative stress obtained from both protein (carbonyl groups) and lipids (thiobarbituric acid-reactive substances) are significantly higher in CD patients. Even in asymptomatic patients, a redox imbalance could influence menstrual and reproductive function. However, there are no follow-up studies assessing these markers in celiac patients. More studies have to be conducted to prove the role of oxidative stress as a culprit in the etiology of CD, including the immunological aspects of gluten-related autoantibodies and infertility.**

For years, GE was discussed in the light of infertility risk. A meta-analysis of observational studies conducted by Lasa et al. revealed a significant association between women diagnosed with infertility and undiagnosed CD patients [OR 3.09 CI 1.74–5.49] [22]. However, when assessing the rate of infertility in patients with diagnosed GE, the OR was 0.99 (95% CI 0/86–1.13). Therefore, the authors concluded that CD is a risk factor for infertility problems when undiagnosed [22]. Other investigators reported patients with primary CD as a cause of unexplained infertility [23,24]. All these cases raised the need for screening for CD in patients with unexplained fertility. On the contrary, Dhalwani et al. questioned fertility problems being related to CD. They estimated that women with CD do not possess a greater risk of clinically significant infertility problems than the general population [25]. A recent systematic review and meta-analysis by Glimberg et al. showed similar results. They estimated that in 0.7% of all studies, women with infertility (*n* = 1617, 11 studies) had biopsy-proven CD, and 0.6% of all women had unexplained infertility. Adding serology testing of women increased the prevalence of CD to up to 1.1% of all women with any form of infertility [26]. Grode et al. (2018) estimated that the prevalence of previously undiagnosed, now biopsy-proven CD in women and men with reproduction issues was 0.45 (%CI 0.12–1.14), and that the total prevalence of already diagnosed and unrecognized cases was 0.63% (% CI 0.29–1.12) [27].

Speaking of serologic testing of CD, the European Society of Paediatric Gastroenterology, Hepatology and Nutrition (ESPGHAN) issued new guidelines for diagnosing CD in 2020 [28]. Based on the accumulated data on gluten enteropathy, ESPGHAN updated their guidelines from 2012 by recommending testing for IgA and IgA antibodies against tissue transglutaminase 2 (anti-tTG IgA). The ESPGHAN advises against using deamidated gliadin peptide antibodies (DGP-IgG/IgA) initially, advising that only if the total IgA is low/undetectable, then an IgG-based test should be considered. HLA DQ2-/DQ8 determination and symptoms are non-mandatory. Patients with confirmed autoimmunity (TGA-IgA/EMA-IgA+) but minimal histological changes (Marsh 0/I) require vigilant monitoring [28]. A meta-analysis by Reiteri et al. (2022) demonstrated that the leading organizations and associations engaged in producing guidelines for the process of CD diagnosis and management have not changed significantly during the last decades: The ESPGHAN (2020), European Society for the Study of Coeliac Disease (ECD) (2019), World Gastroenterology Organization (WGO) (2017), Central Research Institute of Gastroenterology, Russia (2016), National Institute for Health and Care Excellence (NICE) (2015), British Society of Gastroenterology (BSG) (2014), and American College of Gastroenterology (ACG) (2013). Their recommendations are valid for children and adults, and they differ slightly, mainly on the specific management of certain aspects (i.e., neurological manifestations management, refractory CD et al.) [29]. In line with this, our previous research assesses the significance of different autoantibodies in diagnosing and following up the disease by investigating the serum levels of antitissue transglutaminase (anti-tTG IgG, IgM, IgA), anti-deamidated gliadin peptides (anti-DGP IgG, IgM, IgA), anti-actin (AAA), and anti-gliadin antibodies (AGA). However, in our study, anti-DGP antibodies showed the highest diagnostic sensitivity and excellent performance (area under the curve 1.000), followed by AGA and anti-tTG (0.994 and 0.992, respectively) [30]. Other studies also confirmed the diagnostic accuracy of anti-DGP antibodies in celiac disease diagnosis [31]. These rates of accuracy and specificity were confirmed over time, supporting their clinical usefulness.

In respect to reproduction and its association with CD, as we already mentioned above, the main reproductive changes are delayed onset of menstruation or amenorrhea, early menopause, infertility, and during pregnancy, recurrent miscarriage and pregnancy loss, premature birth, intrauterine growth retardation, and low birth weight. Furthermore, CD may also affect male reproduction by altering the sperm motility and morphology and reducing the sexual drive [32]. It was shown that amongst parents of preterm and/or infants that are small for their gestational age, the prevalence of CD is high (0.35 [0.05–1.39%]-1.60% [0.64–3.27%], respectively) [33]. Still, the mechanism of infertility or pregnancy loss in undiagnosed celiac patients is unclear. Probably, both immunological and hormone factors play a role [32].

A recent cross-sectional study by Koshak et al. (2022) revealed some common autoimmune antibodies in unexplained infertility in women. Along with anti-thyroglobulin, anti-thyroid microsomal, beta 2 glycoprotein IgM, and antinuclear antibodies (ANA), anti-gliadin antibodies IgA and anti-tTG IgA (6.7%) were observed in 26.9% of the investigated women [34]. The authors also established an association between the number of pregnancies and anti-tTG IgG and anti-gliadin IgA. Anti-tTG was also associated with the number of ICSI [34]. Di Simone et al. discussed the role of anti-tTG antibodies in damaging trophoblast via apoptosis. The authors performed an in vitro study with anti-tTG antibodies, obtained from celiac parents, and human primary trophoblastic cells, isolated from the placenta. Anti-tTG antibodies showed the ability to bind to trophoblast and to stimulate their apoptosis [35]. This could be one of the mechanisms of how embryo implantation and pregnancy outcomes are impaired due to CD. The levels of anti-tTG antibodies during pregnancy were also associated with reduced fetal weight and birth weight. KIefte-de Jong et al. investigated a cohort of 7046 pregnant women and tested them in the third trimester. Moreover, in the anti-tTG positive group, the reduction in birth weight was two times greater in women that were carriers of HLA-DQ2/8 [13]. Some researchers speculate on the potential pathogenetic mechanisms of anti-tTG antibodies in causing inflammation, thus contributing to infertility. Tersigni et al. demonstrated that CD possesses higher risks for unexplained fertility, recurrent miscarriage, or intrauterine growth restriction by conducting a systematic review and meta-analysis [36]. Additionally, they found that the risk of reproduction failure is higher in people that are not treated with gluten-free diets.

Regarding the mechanisms of CD in infertility, along with the anti-tTG action to hinder the invasiveness of trophoblast [37], these antibodies could damage endometrial angiogenesis, which has been shown in in vitro models with human endometrial cells and in vivo in murine models. Other authors discussed the role of nutrient deficiencies (as we mentioned above as well), such as zinc deficiency (connected to impaired hormone production, secondary amenorrhea, pre-eclampsia), folic acid, etc. [36].

Considering immunological mechanisms, we have to think about autoimmunity. Although CD is not a typical autoimmune disease, it is not only malabsorption but also immune-mediated impairments of physiological processes. Robinson et al. demonstrated that anti-tTG antibodies may directly affect the placenta. Furthermore, tTG is expressed in endometrial cells and stromal and trophoblast placental cells, with increased expression in the late months of pregnancy [38] Other investigators were interested in the role of anti-tTG antibodies in embryo implantation.

It is possible that tTG, expressed in the syncytiotrophoblast, may be the target for the named antibodies, directly influencing the pregnancy outcomes [39,40]. Interestingly, Anhun et al. reported that anti-tTG antibodies IgA bind directly to the syncytial surface of the placenta, leading to the suppression of tTG activity [41]. This could be attributed to the hypothesis explaining a functional placenta development impairment by inhibiting syncytial tTG. Myrsky et al. demonstrated the negative effect of anti-tTG antibodies on the human umbilical vein endothelial cell cytoskeleton. They discussed the probability of anti-tTG antibodies binding to the cell surface and interacting with actin fibers of the cytoskeleton, leading to the disarrangement of the F-actin cytoskeleton [42]. A recent study by Heydari et al. investigated cytokine profiles in treated CD patients. They found that IL-6 and IFNy were increased in celiac patients compared with healthy controls and that there were significant correlations between IFNy levels and abortion, IL-1 and weight loss and infertility in CD patients, and IFNy and abortion and infertility in the non-celiac gluten sensitivity group [43]. To sum up, there are some data on the involvement of celiac-related antibodies in infertility, which we discussed above. However, data so far are not enough to conclude whether a GFD could positively impact the reproductive functions of these people.

## 3. Undiagnosed CD’s Impact on Male Fertility

In contrast to the extensive research focused on female infertility in the context of CD, there is a noticeable absence of evidence concerning the impact of CD on male fertility. The available studies examining the correlation between CD and male infertility primarily aim to ascertain the prevalence of infertility among men with CD and its potential underlying mechanisms. In line with this, a Swedish nationwide population-based study conducted by Zugna et al. [6] demonstrated that CD is associated with an elevated risk of infertility in men, but their study was mainly restricted to women with infertility problems. However, a study by Farthing MJ et al. yielded contrasting results that undiagnosed CD can disrupt the spermatogenesis process, leading to infertility among men [7]. Males afflicted with this condition often experience tissue resistance or insensitivity to androgens. The primary control point for endocrine functions, known as the hypothalamus-pituitary, might undergo an imbalance if there is an elevation in prolactin and follicle-stimulating hormone (FSH). Plasma testosterone and the free testosterone index exhibited elevated levels, while dihydrotestosterone levels were concomitantly reduced. Elevated serum luteinizing hormone levels were also detected. According to a study, hypogonadism, which is characterized by reduced functional activity of the gonads, particularly the testicles, was found in 7% of male CD patients [7].

An in-depth scrutiny of sperm morphology and motility revealed significant anomalies, echoing observations similar to those seen in terms of Crohn’s disease. Additional studies have documented instances of oligospermia [7]. Interestingly, improvements in sperm morphology became apparent after the elimination of dietary gluten [7]. Notably, these hormone levels appeared to return to normalcy as the small intestinal architecture improved under a gluten-free diet.

Incorporating CD screening into the diagnostic protocol for men experiencing infertility is advisable, particularly when standard assessments cannot pinpoint a clear underlying cause. Implementing a strategy for CD screening among high-risk individuals, including those affected by other autoimmune disorders, is a reasonable approach given the heightened prevalence within these specific groups. Managing CD through adherence to a GFD is anticipated to alleviate recognized complications of the condition, thus yielding advantages for both overall health and heightened fertility.

## 4. Pathophysiology of Villous Atrophy in Gluten Enteropathy

The exact pathogenesis of the condition is not clear. However, a hypothesis is that within the lumen of the gastrointestinal tract, the process of gluten/gliadin digestion is only partially completed, resulting in the presence of sizable fragments that are capable of traversing the intestinal barrier. This event subsequently induces an elevation in intestinal permeability, accompanied by the release of zonulin [44]. The presentation of antigens by HLA-DQ2/8 antigen-presenting cells (APCs) triggers an adaptive immune response that is characterized by the secretion of pro-inflammatory cytokines, the mobilization of cytotoxic T cells, and the stimulation of B cells that are specific to gliadin. However, HLA DQ2/8 is a prerequisite but not sufficient factor for CD development.

The role of heredity in CD pathogenesis is also well evaluated, supported by the observation of familial incidence and HLA association of the disease [45]. According to dates provided in the literature, up to 20% of first-degree relatives are affected by the disease, indicating hereditary solid components such as HLA class II genes. HLA DQ2 and HLA DQ8 on chromosome 6 are implicated in the genetic susceptibility. After entering the gut lumen, gliadin (a prolamine part of the gluten) is digested into aminoacids and peptides. However, in some individuals and/or inflammatory conditions, gliadin could penetrate the mucosa and activate intraepithelial lymphocytes. Tissue transglutaminase (tTG) is the enzyme in the mucosa that catalyzes the hydrolysis (deamidation) of aminoacid glutamine into glutamate in glutamine residues of gliadin [46]. It was demonstrated that tTG selectively deamidates gliadin molecules that have entered the lamina propria of the mucosa. The complex of gliading together with tTG forms a neo-antigen., and DGPs, after being processed by tTG, also become neoantigens. Antigens (i.e., gliadin) and neoantigens (i.e., gliadin-tTG complex, DGPs) are then presented via HLA-DQ2 (or DQ8) on the APC for recognition to CD4+ T cells and activate them [46]. It is worth mentioning that this process is not performed in healthy individuals.

This mechanism unconditionally leads to the release of IFNγ and IL21, which leads to epithelial damage. An innate immune response by intraepithelial lymphocytes (IELs), which express NK receptors MHC class I, bind chains A and B and HLE on epithelial cells and lead to the destruction of the epithelium of the small intestine (Marsh classification) (Figure 2 and Figure 3) [47]. This ultimately leads to the activation of specific B cells and the generation of anti-gliadin, anti-tTG, and anti-DGP antibodies. High levels of serum anti-tTG2 Ig Abs and their deposition on the basement membrane of the epithelium of the small intestinal villi and crypts are demonstrated.

In Figure 2 and Figure 3, we present illustrative histopathological changes in the duodenal mucosa in different stages of CD.

The abovementioned activities culminate in the deterioration of mucosal integrity, leading to malabsorption as one of the most critical clinical symptoms. CD-related malabsorption syndrome is an immune-mediated inflammatory process of the small intestine, characterized by impaired intestinal absorption and transport of nutrients and their digestion products through the small intestinal wall, with a frequency varying around 1% of the general population [48]. Disturbances in absorbing minerals, vitamins, proteins, and fats are particularly important, especially in women with infertility.

In Figure 4 we present our histopathological images from a duodenal biopsy obtained from a patient with CD.

A still, peroral small intestinal biopsy is widely used to investigate and diagnose CD. In general, morphological changes in different forms of malabsorption can be observed. The histological picture of the changes in the small intestine is characterized by chronic enteritis with a partial villous atrophy (a mild form of the disease in which villi fuse with each other and thus become shorter and broad) or subtotal villous atrophy (the severe form, in which there is flattening of mucosa due to more advanced villous fusion), with an increased number of lymphocytes and plasma cells in lamina propria (>30 IEL/100 enterocytes represents pathological lymphocytosis).

Goblet cells are increased in number, as is the amount of Paneth cells (which is a good prognostic indicator), because their reduction is evidence of a poor regenerative ability of small intestine mucosa. In cases with subtotal atrophy, goblet cells are few. The basement membrane of the mucous membrane is destroyed [49]. According to meta-analyses found in the literature, which included a total of 23 research studies, including women with complete infertility, women with idiopathic infertility, and women with recurrent miscarriages, the prevalence of serological markers for CD among women with infertility is about 1.3–1.6%, which means that women suffering from such conditions are three times more likely to develop CD. However, due to the small number of cases studied, more studies must be conducted to determine the relationship between CD and fertility disorders accurately [50].

## 5. Summary Table of Meta-Analyses—CD and Infertility

We have conducted a literature research for all patients with CD and infertility problems, and we have summarized our finding in Table 1. This study included all published data in the last decade.

## 6. Dietary Habits and Their Influence on Infertility


*Dietary habits and their impact on female infertility*


Diets enriched with plant-derived proteins have shown a positive potential in enhancing insulin sensitivity and insulin-like growth factor (IGF-I) levels, which are significant factors in metabolic and reproductive health [59]. The impact of carbohydrate diets on blood glucose levels is often evaluated through the glycemic index and glycemic load [59]. A higher consumption of dietary carbohydrates and a higher glycemic load have been linked to an increased incidence of infertility in healthy women [60]. This relationship could be attributed to elevated insulin levels leading to heightened IGF-I and androgen levels, which might give rise to endocrine conditions resembling those associated with polycystic ovarian syndrome [61].

A widely discussed aspect concerning protein products is the potential impact of gluten on fertility. It is important to note that excluding gluten from the diet for individuals without celiac disease offers no discernible benefits [62,63,64]. Significantly, certain food components that are recognized as being detrimental to health are found in higher quantities within a gluten-free diet (GFD) than a standard diet. For instance, a GFD has been associated with elevated dietary exposure to arsenic [65]. Dietary therapies of this type pose a potential risk to fertility, as they frequently result in reduced fiber intake and increased consumption of saturated fatty acids and foods with a higher glycemic index [66]. The positive impact of a high-fiber diet on female reproductive health could be attributed to its ability to regulate blood glucose levels, resulting in a lower dietary glycemic load and index [67]. Slower fermentation of fibers can foster the growth of beneficial gut microbes, contributing to overall health benefits for the host [68]. On the other hand, this can positively impact the overall well-being of female reproductive health and the outcomes associated with assisted reproductive technologies.

Research indicates the significance of fatty acids in early reproductive processes, such as oocyte maturation and embryo implantation [68,69]. Diets that are rich in fatty acids have been associated with enhanced fertility and pregnancy rates, possibly due to their positive influence on steroid and prostaglandin production [70]. Notably, polyunsaturated fatty acid (PUFA) intake has shown promise in enhancing female reproductive parameters, including oocyte quality and embryo implantation [71]. The specific type and quantity of PUFA could directly impact ovarian steroid synthesis, oocyte maturation, and overall pregnancy outcomes, possibly influencing steroidogenesis and prostaglandin synthesis [72,73,74,75].

Vitamins and minerals are pivotal components of the diet, playing roles in various catabolic and anabolic processes that are associated with female reproduction [76]. Folate, in particular, plays a critical role in human reproduction by affecting DNA synthesis, amino acid metabolism, and methionine production [77]. The BioCycle study further supports the link between synthetic folate intake, increased luteal progesterone levels, and a reduced risk of sporadic anovulatory cycles in premenopausal women [78]. However, we should keep in mind the macro- and micronutritient deficiency in pregnant patients on GFD. Furthermore, considering the physiology of embryonic and fetal development, it is crucial to carefully consider the timing and length of gluten avoidance, since it may potentially result in deficits of critical micronutrients like iron and folate. The occurrence of neural tube abnormalities is more likely in pregnant individuals who experience a shortage in folate within the initial four weeks of gestation. Iron insufficiency becomes particularly significant during the third trimester due to the substantial rise in iron requirements resulting from the fast expansion of the placenta, fetus, and maternal blood pool [79].


*Dietary habits and their influence on male infertility*


The impairment in the absorption of crucial elements like zinc, selenium, folic acid, and fat-soluble vitamins finds support in its direct correlation with abnormalities in seminal indicators. Zinc deficiency is associated with manifestations like hypogonadism, reduced sperm counts, and impaired sperm motility, potentially contributing to male infertility [80]. The process of sperm maturation is notably reliant on zinc, with zinc fingers—protein domains rich in histidine and cysteine—playing a pivotal role in DNA-binding proteins. The presence of zinc ions is crucial for maintaining the stability of these structures [81]. Similarly, a selenium deficiency could contribute to abnormalities in the sperm morphology [82]. A critical consequence of a deficiency in vitamin A might be the disruption of Sertoli cell function and early spermatogenesis stages. Vitamin A deficiency can impact the proliferation and function of the epididymal epithelium, along with spermatid maturation [83]. Vitamin E also holds significance, supporting the proper differentiation and function of the epididymal epithelium, spermatid maturation, and the secretion of prostate proteins. Moreover, it improves sperm motility by inhibiting lipid peroxidation [84].

An association between sugar consumption and reduced sperm motility could be attributed to heightened insulin resistance, resulting in limited glucose utilization by sperm [85]. In the context of sperm cells, glucose serves as the primary substrate for glycolysis, a process through which it is converted into pyruvate and/or lactate to generate cellular energy in the form of ATP. Consequently, a decline in sperm’s capacity to uptake and process glucose might lead to decreased ATP levels, which are essential for maintaining optimal sperm motility.

A diet that is low in protein has been identified as a potential risk factor for male-factor infertility, resulting in notable decreases in the weights of the testis, epididymis, and seminal vesicles, along with a reduction in serum testosterone levels [86]. While comprehensive studies concerning non-celiac gluten sensitivity (NCGS) and male fertility disorders are lacking, deficiencies in iron, folic acid, vitamin D, and B12 have been reported in a subset of NCGS patients [87]. There is a possibility that the documented deficiencies in iron, folate, vitamin D, and B12 in NCGS patients could potentially affect male fertility.

## 7. Gluten-Free Diet

It is expected that individuals adhering to a rigorous GFD regimen will exhibit a negative result in an antitissue transglutaminase immunoglobulin A (anti-tTg IgA) test during a period of 6 to 12 months [88]. According to the literature data, after a period of 2 years on a GFD, a majority of 76.2% of patients exhibited a typical endoscopic appearance in the duodenum. In a study by Annibale et al., patients were categorized based on age, and it was observed that patients between the ages of 15 and 60 exhibited a noteworthy improvement within a 12-month period (*p* < 0.0001 for patients aged 15–45 years; *p* < 0.003 for patients aged 46–60 years). However, patients above the age of 60 did not show statistically significant improvement in endoscopic findings, even after 24 months of initiating the gluten-free diet (GFD). A histological examination revealed the presence of “normal” histology in 59.5% of individuals who adhered to a GFD for a duration of 24 months. Significant histological improvement throughout a 12-month period was observed exclusively in the younger patient population, specifically those aged 5 to 30 years (*p* < 0.034) [89,90].

## 8. Conclusions

There is no doubt that untreated CD adversely affects both male and female reproduction. Patients with minimal symptoms also appear to have a significantly higher risk of problems. These complications can cause significant distress for couples and negatively impact the future health of their offspring. According to data in the literature, CD should be suspected in women with menstrual anomalies, infertility, and adverse pregnancy outcomes. All healthcare providers should be aware of these diverse manifestations of the disease. Treating the condition is beneficial and can prevent symptoms and improve the quality of life. Screening for CD (both serological with autoantibodies and histological) should be considered as part of the treatment for patients with unexplained infertility, recurrent spontaneous abortions, and unexplained intrauterine growth retardation. The challenge is to identify women with silent CD and treat them with a GFD and supplements, which can prevent menstrual and other reproductive dysfunction.

## Figures and Tables

**Figure 1 antibodies-12-00079-f001:**
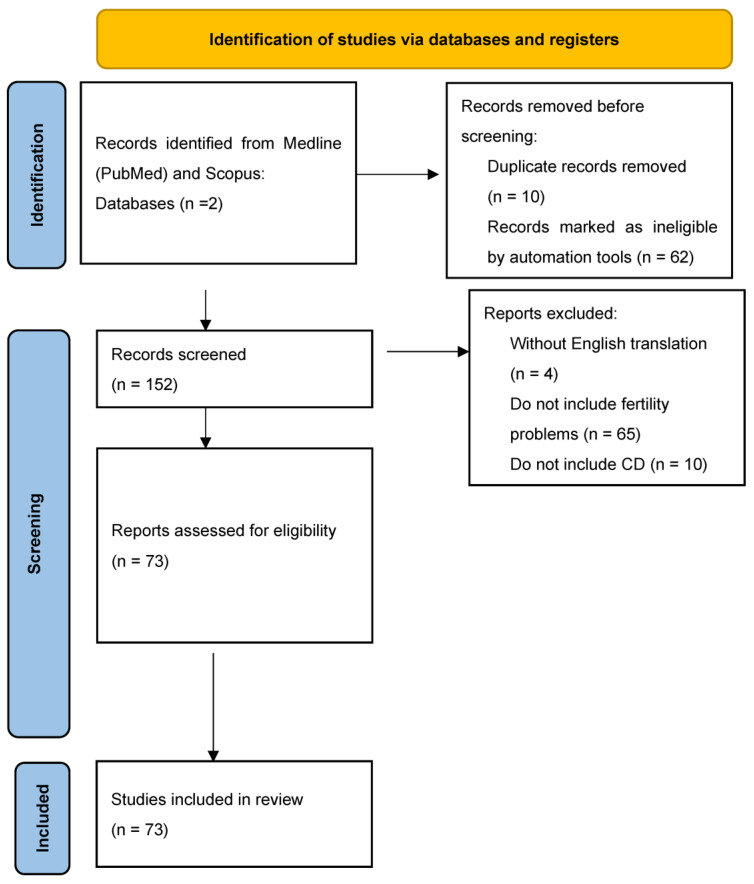
X Prisma chart—search strategy.

**Figure 2 antibodies-12-00079-f002:**
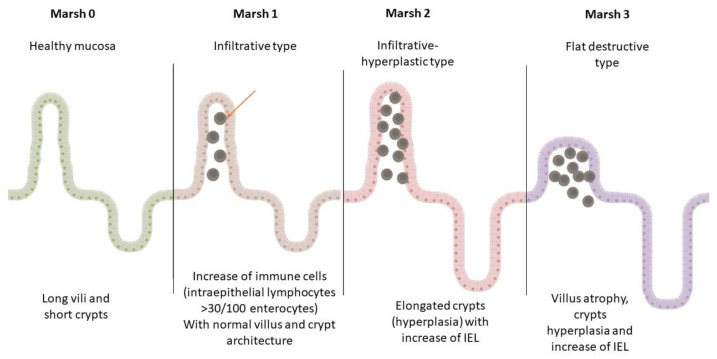
Marsh classification of CD mucosal changes. This figure is created using BioRender.com.

**Figure 3 antibodies-12-00079-f003:**
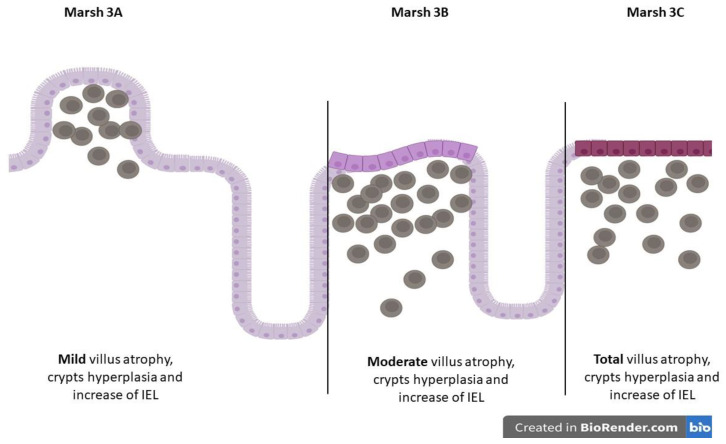
Marsh 3 subclassification of CD mucosal changes. This figure is created using BioRender.com.

**Figure 4 antibodies-12-00079-f004:**
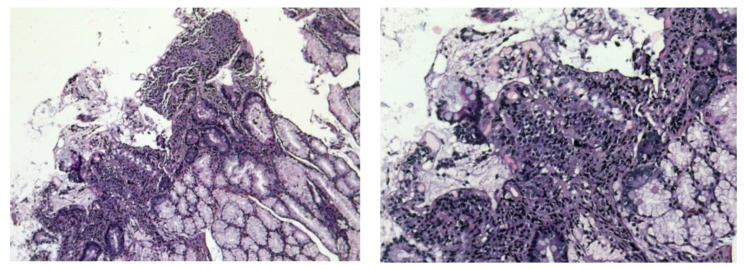
Duodenal mucosa with increased intraepithelial lymphocytes (intraepithelial lymphocytosis), marked villous atrophy, and crypt hyperplasia/Marsh type 3b/HE×50, ×100.

**Table 1 antibodies-12-00079-t001:** Summarized data about CD patients with infertility problems.

Author	Year	№ of Patients	Gender	Mean Age ± SD	Evaluated Factor	Serologic CD
Kutteh et al. [51]	2019	708	F	33 ± 5.8	RM—708	TTG IgA—9; EMA—6; DGPA—4
Grode et al. [27]	2019	885	F—453 M—432	31.9 ± 5	Infertility—884 RM—1	TTG IgA/ DGPA—8
Juneau et al. [52]	2018	995	F	35.9 ± 4.0	Infertility—995	TTG IgA—24; EMA—22; DGPA—NR
Gunn et al. [53]	2018	393	F	35.7 ± 4.3	Infertility—393	TTG IgA—1; EMA—NR; DGPA—NR; Biopsy—NR
Sabzebari et al. [54]	2017	100	F/M	NR	Infertility—100	TTG IgA—14; EMA—NR; DGPA—NR
Sarikaya et al. [55]	2017	45	F	28	RM—45	TTG IgA—1; EMA—NR; DGPA—1
Karaca et al. [56]	2015	65 couples	F/M	33.40 ± 4.59	Infertility—65	TTG IgA—1 male; EMA—1 male; DGPA—1 male
Machado et al. [57]	2013	170	F	35 ± 6	Infertility—65	TTG IgA—6; EMA—3; DGPA—NR
Sharshiner et al. [58]	2013	116	F	30.16 ± 4.43	RM—116	TTG IgA—1; EMA—0; DGPA—0

TTG—tissue transglutaminase; EMA—endomisial antibody; DGPA—deamidated gliadin peptides; CD—celiac disease; RM—recurrent misabortion; NR—not reported. F—female; M—male.

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
