# Peer review of "Navigating the Challenges of Gluten Enteropathy and Infertility: The Role of Celiac-Related Antibodies and Dietary Changes"

_2073-4468, 2023, doi:10.3390/antib12040079_

Round 1

Reviewer 1 Report

Comments and Suggestions for Authors

This paper reviews the relationship between celiac disease and infertility. The title was "Role of celiac-related antibodies," but the review did not focus on the important role of antibodies in infertility diagnosis and prognosis. Some changes are suggested to make the review conform to the title.

1. Please discuss the causes and possible mechanisms of neonatal low body weight caused by anti-tTG antibodies in pregnant women (Line 337-339).

2. Please explain the effect of a gluten-free diet on the negative turning of anti-tTG antibodies and how long it takes for autoantibody seroconversion after the diet change.

3. What is the effect of a gluten-free diet on macro and micronutrient intake in pregnant women? Does it affect the nutritional status of the pregnant woman and the fetus?

Author Response

Name of the journal: Antibodies

Title: Navigating the Challenges of Gluten Enteropathy and Infertility: role of celiac-related antibodies

Authors: Monika Peshevska-Sekulovska1,2, Milena Gulinac 3, Radoslav Rangelov 4, Desislava Docheva 4, Tsvetelina Velikova 1, Metodija Sekulovski1,4,5*

Dear Editor,

Dear reviewers,

Thank you for your time to revise our manuscript: Navigating the Challenges of Gluten Enteropathy and Infertility: role of celiac-related antibodies, Authors: Monika Peshevska-Sekulovska, Milena Gulinac, Radoslav Rangelov, Desislava Docheva, Tsvetelina Velikova, Metodija Sekulovski.  We have incorporated all the suggestions made by the reviewers. Those changes are highlighted within the manuscript. Please see below, in blue, for a point-by-point response to the reviewers’ comments. All page numbers refer to the revised manuscript file with tracked changes.

Reviewer 1

This paper reviews the relationship between celiac disease and infertility. The title was "Role of celiac-related antibodies," but the review did not focus on the important role of antibodies in infertility diagnosis and prognosis. Some changes are suggested to make the review conform to the title.

  1. Please discuss the causes and possible mechanisms of neonatal low body weight caused by anti-tTG antibodies in pregnant women (Line 337-339).

  • Thank you for the suggestion. We revised the text accordingly.

  1. Please explain the effect of a gluten-free diet on the negative turning of anti-tTG antibodies and how long it takes for autoantibody seroconversion after the diet change.
  • Thank you very much for the time to review our paper and for the chance to revise it. We acknowledge that our paper might have some issues in conformity with your critical points. We have revised the paper accordingly and we have added this important part in newly formed section GFD.

  1. What is the effect of a gluten-free diet on macro and micronutrient intake in pregnant women? Does it affect the nutritional status of the pregnant woman and the fetus?
  • Thank you for the suggestions. We have implemented them in the end of the part with Dietary habits and their impact on female infertility.

Reviewer 2 Report

Comments and Suggestions for Authors

I should first thank for inviting me as potential reviewer to read and comment on paper entitled ‘Navigating the Challenges of Gluten Enteropathy and Infertil- 2 ity: role of celiac-related antibodies’’.

In the current study, the authors aimed to to conduct a comprehensive 65 review of existing literature pertaining to this specific topic and to elucidate the role of the 66 autoantibodies in its pathogenesis.

The main title accurately reflects the major topic and content of the study.

The abstract summarizes and reflects the work described in the manuscript. Also, the abstract presents the significant points related to the background.

The section of the manuscript is well organized. The conclusions are drawn appropriately supported by the literature. The manuscript adequately describes the background, present status and significance of the study. The manuscript interprets the findings adequately and appropriately, highlighting the key points clearly. However, the authors made few references to recently published articles.

I think that it will contribute to the literature. I have some minor crticisms.

-       The manuscript appropriately cites the important and authoritative references but does not cite the recent published articles in the Introduction part of the manuscript. If the recent published article about celiac disease are cited, the manuscript would be better.

-       The authors does not cite the recent published articles in the part of the ‘’Pathophysiology of villous atrophy in gluten enteropathy’’. The authors cited ‘’Hospital (Rio J). 1966 May;69(5):1009-27’’. If the recent published articles about frequency of celiac disease in siblings are cited, the manuscript would be better. The aurhors can be benefit fom the following article.

In the recent study called ‘’ Frequency of celiac disease and distribution of HLA-DQ2/DQ8 haplotypes among siblings of children with celiac disease’’ (World J Clin Pediatr 2022;11:351-59), the authors found that the prevalence of CD was found to be 10.7% in siblings of celiac patients. In addition to that, one-third of the siblings diagnosed with CD were asymptomatic. Also, the authors detected HLA-DQ alleles in 98.2% of celiac patients and 100% in siblings diagnosed with CD.

Author Response

Reviewer 2

I should first thank for inviting me as potential reviewer to read and comment on paper entitled ‘Navigating the Challenges of Gluten Enteropathy and Infertil- 2 ity: role of celiac-related antibodies’’.

In the current study, the authors aimed to to conduct a comprehensive 65 review of existing literature pertaining to this specific topic and to elucidate the role of the 66 autoantibodies in its pathogenesis.

The main title accurately reflects the major topic and content of the study.

The abstract summarizes and reflects the work described in the manuscript. Also, the abstract presents the significant points related to the background.

The section of the manuscript is well organized. The conclusions are drawn appropriately supported by the literature. The manuscript adequately describes the background, present status and significance of the study. The manuscript interprets the findings adequately and appropriately, highlighting the key points clearly. However, the authors made few references to recently published articles.

  • Thank you for the critical notes. We included more recent references on the topic based on your recommendations. We believe that with the revision the paper has been improved significantly.

I think that it will contribute to the literature. I have some minor crticisms.

-       The manuscript appropriately cites the important and authoritative references but does not cite the recent published articles in the Introduction part of the manuscript. If the recent published article about celiac disease are cited, the manuscript would be better.

-       The authors does not cite the recent published articles in the part of the ‘’Pathophysiology of villous atrophy in gluten enteropathy’’. The authors cited ‘’Hospital (Rio J). 1966 May;69(5):1009-27’’. If the recent published articles about frequency of celiac disease in siblings are cited, the manuscript would be better. The aurhors can be benefit fom the following article.

In the recent study called ‘’ Frequency of celiac disease and distribution of HLA-DQ2/DQ8 haplotypes among siblings of children with celiac disease’’ (World J Clin Pediatr 2022;11:351-59), the authors found that the prevalence of CD was found to be 10.7% in siblings of celiac patients. In addition to that, one-third of the siblings diagnosed with CD were asymptomatic. Also, the authors detected HLA-DQ alleles in 98.2% of celiac patients and 100% in siblings diagnosed with CD.

  • Thank you very much for the time to review our paper and for the chance to revise it. We acknowledge that our paper might have some issues in conformity with your critical points. We have revised the paper accordingly.We have add the recommended references.

Reviewer 3 Report

Comments and Suggestions for Authors

General Statement

This manuscript is a review on the role of gluten enteropathy and antibodies related to infertility. It reviews publications from the last 10 years and, based on these studies, explains the data and controversies present in these publications. There are certain aspects that need improvement.

Main comments

In section 5, in the explanation of the pathophysiology of celiac disease, there is no mention of the role of transglutaminase, the deamidation of gliadin peptides, and the relevance of this post-translational modification for presentation by HLA-DQ2/DQ8 molecules. In my opinion this is an important aspect that is not well-reflected.

Minor comments

The bibliographic references should be placed before the period at the end of the sentence to avoid confusion.

Point 3 of the review, despite providing information on other causes of infertility, is not the focus of the review and should be omitted.

In section 6, when referring to relevant autoantibodies for the diagnosis of celiac disease, reference is made to ESPGHAN, but in the majority of cases, fertility problems are a clinical issue in adulthood, and no guidelines are referenced for the diagnosis of this condition in that age group.

In this same section, the results of a study on the performance of different autoantibodies related to celiac disease are presented, which adds little value to the review's objective.

In line 340, there is an error; "DQ2/9" should be replaced with "DQ2/8."

In lines 373-375, it is mentioned that there is limited data to understand the influence of the gluten-free diet (GFD) on fertility, and there is no mention that the GFD reduces or eliminates autoantibody levels. This could be the reason why the GFD improves fertility in patients with celiac disease.

Line 376 actually corresponds to point 7, not point 6.

Author Response

Reviewer 3

General Statement

This manuscript is a review on the role of gluten enteropathy and antibodies related to infertility. It reviews publications from the last 10 years and, based on these studies, explains the data and controversies present in these publications. There are certain aspects that need improvement.

Main comments

In section 5, in the explanation of the pathophysiology of celiac disease, there is no mention of the role of transglutaminase, the deamidation of gliadin peptides, and the relevance of this post-translational modification for presentation by HLA-DQ2/DQ8 molecules. In my opinion this is an important aspect that is not well-reflected.

  • Thank you very much for the critical note, we agree completely. We have added a brief paragraph on the role of the enzyme and the result of the deamidation with focus on the genetic background and immune consequences.

Minor comments

The bibliographic references should be placed before the period at the end of the sentence to avoid confusion.

  • Thank you for your note, we revised the paper accordingly.

Point 3 of the review, despite providing information on other causes of infertility, is not the focus of the review and should be omitted.

  • We agree with the referee that this is not the topic of our work, however, we believe this is an essential part of the paper, since it is related to the gluten-free diet. We can propose to add this in the title - Navigating the Challenges of Gluten Enteropathy and Infertility: role of celiac-related antibodies and dietary factors

In section 6, when referring to relevant autoantibodies for the diagnosis of celiac disease, reference is made to ESPGHAN, but in the majority of cases, fertility problems are a clinical issue in adulthood, and no guidelines are referenced for the diagnosis of this condition in that age group.

  • The referee is right to point out that ESPGHAN is the pediatric organization. We searched again for recommendations for adults. We found a meta-analysis thet covers the main organizations: European Society for the Study of Coeliac Disease (ECD) (2019), World Gastroenterology Organization (WGO) (2017), Central Research Institute of Gastroenterology, Russia (2016), National Institute for Health and Care Excellence (NICE)(2015), British Society of Gastroenterology (BSG), (2014), and America College of Gastroenterology (ACG)(2013).

In this same section, the results of a study on the performance of different autoantibodies related to celiac disease are presented, which adds little value to the review's objective.

  • We acknowledge the note and we reduce the information. However, we would appreciate if the referee agree to keep this statement because the rate of these antibodies is important for the overall diagnostic accuracy of celiacdisease, and this should be emphasized. We add a sentence to explain why this is important information.

In line 340, there is an error; "DQ2/9" should be replaced with "DQ2/8."

  • Thank you for your note. We have corrected our mistake.

In lines 373-375, it is mentioned that there is limited data to understand the influence of the gluten-free diet (GFD) on fertility, and there is no mention that the GFD reduces or eliminates autoantibody levels. This could be the reason why the GFD improves fertility in patients with celiac disease.

  • Thank you very much for the valuable note. We have added the information in section GFD

Line 376 actually corresponds to point 7, not point 6.

  • Thank you very much for the valuable note. We have corrected the numbering of the sections.

Reviewer 4 Report

Comments and Suggestions for Authors

I have read with interest the manuscript entitled: “Navigating the Challenges of Gluten Enteropathy and Infertility: role of celiac-related antibodies. 

The paper presents an interesting topic; however, some structure is required to present main findings in an unbiased form. 

Major concerns are the probability of bias due to the conduction of the review, without transparency on how information was added. It is suggested that for proper standardization, authors could follow the extended PRISMA guidelines for focused reviews. 

Tricco, AC, Lillie, E, Zarin, W, O'Brien, KK, Colquhoun, H, Levac, D, Moher, D, Peters, MD, Horsley, T, Weeks, L, Hempel, S et al. PRISMA extension for scoping reviews (PRISMA-ScR): checklist and explanation. Ann Intern Med. 2018,169(7):467-473. doi:10.7326/M18-0850.

Furthermore, the paper lacks a proper organization, sections 2-6 (inserted twice) should be considered into a single section identified as results. Section 3 could be a discussion, here authors can provide a summary of the evidence together with limitations of the review and their conclusions. 

The abstract lacks a statement related to the purpose of the review; however, in the introduction section authors state that their objective was “to conduct a comprehensive review of existing literature pertaining to his specific topic and to elucidate the role of the autoantibodies in its pathogenesis”. Authors should provide a clearer and specific objective, both in the abstract as well as the introduction section. 

Here are minor concerns for different sections:

1.       The introduction section of the manuscript is too broad, and it fails to provide a comprehensive description of the main concepts that will be presented later in the manuscript. Important concepts and definitions are missing as well as a rationale for the review. Adding questions to this section might be useful, to clarify the purpose. In accordance with the title some important information that should be included in this section should be related to the origins of infertility in CD and the participation of autoantibodies production on it. The first is rather described in section 2 and the last is entirely missing from the introduction section. 

2.       Authors should provide an introductory paragraph on why the dietary habits and their influence on infertility are provided in this paper, this information seems to be completely apart from the objective of the paper. 

3.       Pag. 3, line 122 what does PCOS stand for? It is recommended that when abbreviations are presented for the first time they can be defined. 

4.       The first paragraph from section 4 could be added to the introduction section. 

5.       On section 4, authors should provide a possible explanation to inconsistencies reported on the prevalence of infertility in men with CD (references 6 and 7).

6.       Section 5 (except for the las part of the final paragraph) seems to be, unrelated to the objective of the review; nevertheless, it contains important information that could be part of the introduction section.  

7.       Page 7 line 284, there seems to be an error in the 95% CI reported.  

8.       Number 6 is repeated in two sections.

9.       Table 1 could be presented in a horizontal form instead of vertical, that would make it easier to read. But more importantly, what is the purpose of this table? To present serologic findings? What does RM stand for? What information can be acquired from this table?

10.    Consider addressing the benefits from treatment into the decreasing prevalence of infertility among CD patients. 

Author Response

Reviewer 4

I have read with interest the manuscript entitled: “Navigating the Challenges of Gluten Enteropathy and Infertility: role of celiac-related antibodies. 

The paper presents an interesting topic; however, some structure is required to present main findings in an unbiased form. 

Major concerns are the probability of bias due to the conduction of the review, without transparency on how information was added. It is suggested that for proper standardization, authors could follow the extended PRISMA guidelines for focused reviews. 

Tricco, AC, Lillie, E, Zarin, W, O'Brien, KK, Colquhoun, H, Levac, D, Moher, D, Peters, MD, Horsley, T, Weeks, L, Hempel, S et al. PRISMA extension for scoping reviews (PRISMA-ScR): checklist and explanation. Ann Intern Med. 2018,169(7):467-473. doi:10.7326/M18-0850.

  • Thank you for the valuable recommendations, we provided this information and believe that this makes the process of selecting paper for the review transparent.

Furthermore, the paper lacks a proper organization, sections 2-6 (inserted twice) should be considered into a single section identified as results. Section 3 could be a discussion, here authors can provide a summary of the evidence together with limitations of the review and their conclusions. 

  • We agree with the referee that the structure and comprehension of the review could be improved. Thus, we have incorporated the section 2 and 6 in one section and also, we have put section 3 with GFD in the discussion part.

The abstract lacks a statement related to the purpose of the review; however, in the introduction section authors state that their objective was “to conduct a comprehensive review of existing literature pertaining to his specific topic and to elucidate the role of the autoantibodies in its pathogenesis”. Authors should provide a clearer and specific objective, both in the abstract as well as the introduction section. 

  • The refereee is absolutely right to point this out. Thus, we have revised the abstract

Here are minor concerns for different sections:

  1. The introduction section of the manuscript is too broad, and it fails to provide a comprehensive description of the main concepts that will be presented later in the manuscript. Important concepts and definitions are missing as well as a rationale for the review. Adding questions to this section might be useful, to clarify the purpose. In accordance with the title some important information that should be included in this section should be related to the origins of infertility in CD and the participation of autoantibodies production on it. The first is rather described in section 2 and the last is entirely missing from the introduction section. 
  • We agree with the referee that the structure and comprehension of the introduction part could be improved. We revised that part

  1. 3, line 122 what does PCOS stand for? It is recommended that when abbreviations are presented for the first time they can be defined. 
  • Thank you for your note. We have corrected our mistake.

  1. The first paragraph from section 4 could be added to the introduction section. 
  • Thank you very much for the valuable note. We have move that part in the introduction.

  1. On section 4, authors should provide a possible explanation to inconsistencies reported on the prevalence of infertility in men with CD (references 6 and 7).
  • We added all the needed information.

  1. Section 5 (except for the las part of the final paragraph) seems to be, unrelated to the objective of the review; nevertheless, it contains important information that could be part of the introduction section. 
  • Thank you very much for the critical note. We can understand the point of the referee. However, we would be grateful if the referee accepts our idea to leave the information in section 5 for two reasons: 1. Introduction is already prepared to include background and hypothesis, and moving information there would damage the structure of the introduction; 2. We believe that this information is critical for supporting the ideas in section 5.

  1. Page 7 line 284, there seems to be an error in the 95% CI reported.  
  • The referee is absolutely right to point this out. Therefore, we have corrected this mistake.

  1. Number 6 is repeated in two sections.
  • Thank you very much for the valuable note. We have corrected the section numbering.

  1. Table 1 could be presented in a horizontal form instead of vertical, that would make it easier to read. But more importantly, what is the purpose of this table? To present serologic findings? What does RM stand for? What information can be acquired from this table?
  • Thank you for the further questions. The table included in our article, tends to present the summarized literature data, that has been published so far in respect to patients with reproductive problems and CD.
  • Also, we agree that the auditoria would benefit from horizontal form of the table. Thus, we have correctedFurthermore, RM stands for recurrent mis abortion and we have inserted the abbreviation below the table.

  1. Consider addressing the benefits from treatment into the decreasing prevalence of infertility among CD patients. 
  • Thank you for the critical point. We have addressed the benefit from GFD in section 7.

Round 2

Reviewer 3 Report

Comments and Suggestions for Authors

The current revision of the manuscript has improved upon the previous version, clarifying certain concepts. Nevertheless, there are still minor errors that need to be addressed.

In line 146, "Mothers bearing the HLA-DQ2 or -DQ8 molecule exhibited a 2-fold increase in the impact of intermediate anti-tTG levels on birth weight." This statement contains relevant information but lacks proper referencing. The reference number 13 does not appear to contain this information; however, it can be correctly attributed to reference 37.

In line 157, it is stated, "Furthermore, it was shown that the human placenta exhibited the presence of anti-tTG expression." The intended meaning is the expression of tTG (tissue transglutaminase) rather than anti-tTG antibodies.

In line 428, the abbreviation DGP is defined, but it has already been used earlier in the text.

Author Response

Dear Editor,

Thank you for your time and effort for reviewing our manuscript: Navigating the Challenges of Gluten Enteropathy and Infertility: role of celiac-related antibodies and dietary changes.

Please find below point-by-point answers to your suggestions.

The current revision of the manuscript has improved upon the previous version, clarifying certain concepts. Nevertheless, there are still minor errors that need to be addressed. In line 146, "Mothers bearing the HLA-DQ2 or -DQ8 molecule exhibited a 2-fold increase in the impact of intermediate anti-tTG levels on birth weight." This statement contains relevant information but lacks proper referencing. The reference number 13 does not appear to contain this information; however, it can be correctly attributed to reference 37.

Dear Editor, we completely agree with you. Thus, we have corrected our mistake.

In line 157, it is stated, "Furthermore, it was shown that the human placenta exhibited the presence of anti-tTG expression." The intended meaning is the expression of tTG (tissue transglutaminase) rather than anti-tTG antibodies.

In line 428, the abbreviation DGP is defined, but it has already been used earlier in the text.

Thank you for you note. We have corrected it.

Reviewer 4 Report

Comments and Suggestions for Authors

Authors have complied with all recommendations, or provide an argument to favor their original presentation of information. 

Author Response

Authors have complied with all recommendations, or provide an argument to favor their original presentation of information.

Thank you for your time and effort for reviewing our manuscript: Navigating the Challenges of Gluten Enteropathy and Infertility: role of celiac-related antibodies and dietary changes.